# EVs from BALF—Mediators of Inflammation and Potential Biomarkers in Lung Diseases

**DOI:** 10.3390/ijms22073651

**Published:** 2021-04-01

**Authors:** Lukasz Zareba, Jacek Szymanski, Zuzanna Homoncik, Malgorzata Czystowska-Kuzmicz

**Affiliations:** 1Chair and Department of Biochemistry, Medical University of Warsaw, Banacha 1, 02-097 Warsaw, Poland; s068791@student.wum.edu.pl (L.Z.); jacek.szymanski3.stud@pw.edu.pl (J.S.); homoncik.zuzanna@gmail.com (Z.H.); 2Faculty of Chemistry, Warsaw University of Technology, Noakowskiego 3, 00-664 Warsaw, Poland

**Keywords:** extracellular vesicles (EVs), bronchoalveolar lavage fluid (BALF), inflammatory lung diseases, lung cancer, exosomes, biomarker

## Abstract

Extracellular vesicles (EVs) have been identified as key messengers of intracellular communication in health and disease, including the lung. EVs that can be found in bronchoalveolar lavage fluid (BALF) are released by multiple cells of the airways including bronchial epithelial cells, endothelial cells, alveolar macrophages, and other immune cells, and they have been shown to mediate proinflammatory signals in many inflammatory lung diseases. They transfer complex molecular cargo, including proteins, cytokines, lipids, and nucleic acids such as microRNA, between structural cells such as pulmonary epithelial cells and innate immune cells such as alveolar macrophages, shaping mutually their functions and affecting the alveolar microenvironment homeostasis. Here, we discuss this distinct molecular cargo of BALF-EVs in the context of inducing and propagating inflammatory responses in particular acute and chronic lung disorders. We present different identified cellular interactions in the inflammatory lung via EVs and their role in lung pathogenesis. We also summarize the latest studies on the potential use of BALF-EVs as diagnostic and prognostic biomarkers of lung diseases, especially of lung cancer.

## 1. Introduction

### 1.1. General Information about EVs

Extracellular vesicles (EVs) are heterogenous, nanoscale membrane vesicles, released by cells for intercellular communication. EV cargo, which may consist of various molecules (proteins, nucleic acids (e.g., DNA, messenger RNA (mRNA), microRNA (miRNA)), or metabolites), is surrounded by a phospholipid bilayer protecting it from degradation. Carrying various signal molecules or genetic information, EVs can initiate or modulate specific physiological or pathological processes [1]. Depending on the biogenesis, EVs are divided into three main subpopulations: exosomes, microvesicles (MVs), and apoptotic bodies [2]. Exosomes are the smallest vesicles in the range of 40–120 nm [3]. Their biogenesis begins with ILV (intraluminal vesicle) formation through inward budding of the endosomal membrane, which causes endosome transformation into a multivesicular body (MVB). ILVs are released from the cell as exosomes through fusion of MVBs with the cellular membrane [4]. In contrast to exosomes, microvesicles (50–1000 nm) are generated by direct outward budding of the plasma membrane. Apoptotic bodies, with the largest size of 50–2000 nm, are created through cell fragmentation during cellular programmed death called apoptosis [5]. Extracellular vesicles are released by all cell types; therefore, they occur in various body fluids, e.g., blood [6], urine [7], or BALF [8]. In spite of the well-defined origin of exosomes and MVs, the current isolation and characterization techniques cannot clearly distinguish between the different types of EVs [9], and there is still huge confusion and imprecision in the nomenclature [10]. Many researchers use the terms exosomes or MVs arbitrarily and not precisely. Therefore, throughout this review, we use the term EVs as a general term for all types of vesicles in the extracellular space, according to the latest Minimum Information for the Study of Extracellular Vesicles (MISEV) 2018 recommendations [11]. However, we keep the original nomenclature of the authors in the discussed references when they were specially identified in the study.

### 1.2. BALF

BALF is a fluid collected during bronchoalveolar lavage performed by flexible bronchoscopy. It is an important diagnostic material that provides information about immunological, infectious, and inflammatory processes taking place in pulmonary alveoli [12]. BALF is enriched with various cellular and noncellular content including epithelial cells, macrophages, cytokines, extracellular vesicles, and many more [8,13]. Since it allows obtaining the cellular and noncellular content from distal airways and lung alveoli, it represents an ideal “lung liquid biopsy” and is now under extensive research as a source of biomarkers for different lung diseases, e.g., for lung cancer [14]. BALF contains a substantial amount of different EVs, although in a lower concentration than in other “natural” biological fluids. Rodriguez et al. detected between 1.8 and 3.8 × 10^8^ EV-like particles by nanoparticle tracking analysis (NTA) in comparison to 4.0–9.8 × 10^8^ particles per mL in plasma [15]. In our own unpublished studies, we also detected higher particle concentrations in plasma than in BALF. However, by fluorescence labeling against exosomal tetraspanins, we could show that the majority of detected particles in BALF are “true” EVs, whereas, in plasma, most particles represent non-EV structures such as lipoproteins (unpublished results). Therefore, BALF may be a better source of vesicular biomarkers than plasma or serum for different lung diseases, all the more so because it may better reflect the tumor microenvironment than the systemic circulation. An example is the usage of BALF-EVs as a diagnostic biomarker in lung cancer. Hur et al. showed that BALF-EVs containing DNA and used for epidermal growth factor receptor (EGFR) genotyping had higher accordance with conventional tissue biopsy compared to plasma EVs. The detection sensitivity of plasma-EV DNA was only 55% in contrast to 100% of BALF-EV DNA [16]. Another type of biomarker that is recently under intensive research is miRNA. Since miRNAs in blood are not only associated with EVs but also proteins such as Argonaute complexes [17] or lipoprotein particles [18], and since currently available isolation methods do not ensure complete purification of plasma-derived EVs from this impurities, lipoprotein-devoid and protein-pure biofluids such as BALF may be better sources of EV-associated miRNAs as biomarkers, additionally providing local sampling [19].

A detailed analysis of the EV subpopulations in BALF for most lung diseases is still lacking; however, one can assume that the EV composition in BALF more or less reflects its cellular composition, which is largely dependent on the disease. In health, lung epithelial cells are considered to be major sources of pulmonary EVs, which contain different membrane-tethered mucins and regulate the normal airway biology including homeostasis and innate defense [20]. Macrophages are also major sources of pulmonary EVs, that normally maintain homeostasis and immune cell production. In addition to these two main cellular EV sources, fibroblasts, mesenchymal stem cells (MSCs), endothelial cells, neutrophils, eosinophils, and other immune cells may contribute in different amounts to the BALF-EV pool, depending on the condition [21,22]. The BALF-EV composition may change dramatically in disease. For example, single vesicle flow cytometry phenotyping showed that 80% of EVs in BALF from healthy mice were secreted from the airway-lining epithelium; however, after induction of allergic airway inflammation by using ovalbumin (OVA), the number of immune cell-derived EVs including miR-223 and miR-142a increased by more than twofold in BALF from allergen-treated mice [23].

EV isolation from BALF is still at an early stage of development; thus, there are only a few established methods that are suited for this purpose. Vesicle integrity, relevant EV population, recovery, and purity of EVs isolates are taken into account during the choice of the appropriate isolation method. In comparison to blood, BALF contains less protein. Moreover, similar to urine and cell culture medium, lipoprotein and chylomicron contamination is excluded. Consequently, isolating EVs from BALF may be more clinically significant [24,25]. However, some BALF samples require dithiothreitol (DTT) incubation for mucus dissolution. Additionally, the bronchoscopy procedure is invasive; thus, access to the material is limited [26,27]. Due to the large volume of BALF and low EV concentration, the most suitable and reliable method is differential centrifugation, which is based on the constituent density and size. Several centrifugation steps at increasing speeds (300–2000× *g*) are applied to remove cells and sizeable debris. Then, larger vesicles and residual debris are removed by further ultracentrifugation (UC) steps using forces from 100,000–200,000× *g* and, finally, the EVs pellet is yielded. The method is relatively simple and reduces contamination risks and costs, despite the necessary equipment for UC. On the other hand, UC is time-consuming and EVs can be contaminated by particles of similar size and density. Moreover, in spite of larger amounts of EVs that can be obtained by UC pelleting, high speeds during UC may damage the EVs. Furthermore, to obtain purer samples with lower protein content or to isolate particular EVs, more developed UC methods are necessary, e.g., density-gradient UC, moving-zone density-gradient, or a combination with other EV isolation methods, e.g., immunomagnetic isolation. Nevertheless, BALF-EV isolation can also be conducted by size-exclusion chromatography (SEC), polymer precipitation, or immunoprecipitation. SEC is a size-based macromolecule separation technique, which allows EV isolation without most lipoproteins and plasma proteins. The advantages of this method are a small sample volume and the obtainment of protein-depleted, non-aggregated, and biologically active EVs. Nonetheless, additional UC steps are required to exclude similar-sized vesicles, e.g., lipoproteins. The usually larger volumes of BALF require sample concentration or multiple columns for SEC, which elongates the duration of isolation. The other techniques are less popular. During the precipitation method, water-excluding polymers are used for mildly modifying the solubility or dispersibility of EVs. Actually, only few groups perform BALF-EV isolation; thus, further investigations are essential [24,25].

Recent studies revealed the potential role of EVs in mediating the inflammation process in the respiratory system [26,27]. Therefore, in this article, we aim to review the current knowledge of EV contribution to inflammation in the pathogenesis of various lung diseases and their potential role as biomarkers. In Section 2, we elaborate on the proinflammatory effects of EVs in lung diseases such as sarcoidosis, chronic obstructive pulmonary disease, cystic fibrosis, asthma, acute respiratory distress syndrome/acute lung injury, interstitial lung disease, allergic inflammation, and lung cancer. Afterward, we discuss the anti-inflammatory effect of EVs. Section 3 collects evidence for the potential role of EVs as biomarkers in lung inflammatory diseases and lung cancer.

## 2. BALF-EVs Modulate Inflammation in Pulmonary Diseases

### 2.1. BALF-EVs Mediate Proinflammatory Effects

#### 2.1.1. Sarcoidosis

Sarcoidosis is an inflammatory systemic disease with difficult diagnostics and only symptomatic treatment. Typically, the formation of sarcoid granulomas is provoked by T cells releasing interferon-γ (IFNγ). Resulting inflammation and tissue damage predominantly affects the lungs and variably affects multiple other organs such as the liver, skin, eyes, heart, and central nervous system. Most patients enter remission; however, in 20% of sarcoidosis cases, pulmonary fibrosis develops, which can lead to many serious complications such as pulmonary hypertension or chronic aspergillus disease and other life-threatening symptoms [28]. The ethology of sarcoidosis is unknown, but some research shows the possible pathogenesis of sarcoidosis involving a connection among human leukocyte antigens (HLA) class II molecules, antigens, and T-cell receptors [29].

Quantitative analysis of exosomes isolated from BALF showed increased expression of immunostimulatory molecules, such as major histocompatibility complex (MHC) class I and II and the tetraspanins cluster of differentiation 9 (CD9), CD63, and CD81 in sarcoidosis patients compared to healthy controls. Moreover, a positive correlation between tetraspanins and HLA-DR was observed [30], which can be explained by the relationship between MHC-II and some tetraspanins as interaction partners [31]. Interestingly, the level of neuregulin-1 (NRG1), a protein involved in cancer progression [32], was also upregulated in patients with sarcoidosis [30]. The role of NRG-1 in sarcoidosis is not clarified, but it may be associated with cell survival and proliferation. Furthermore, in vitro experiments showed that BALF exosomes induced the production of IFN-γ, interleukin-13 (IL-13) in autologous peripheral blood mononuclear cells (PBMCs), and IL-8 in epithetical cells, which proclaimed their role in activation of innate and adaptive immune systems [30].

Further research showed that BALF exosomes stimulate the production of IL-1β, IL-6, and tumor necrosis factor α (TNF- α) from PBMCs and enriched monocytes in similar levels, suggesting a direct impact on monocytes. For some patients, the potent chemotactic agent, C–C motif chemokine ligand 2 (CCL2) was also increased in PBMCs by exosomes. Interestingly, the exosome-driven increase of CCL2 was subsequently reduced by addition of a cysteinyl leukotriene receptor antagonist—montelukast [33], used as an asthma drug [34]. The abovementioned research showed possible roles of exosomes as mediators in inflammation progression and a potential target in sarcoidosis treatment.

#### 2.1.2. Chronic Obstructive Pulmonary Disease

Chronic pulmonary obstructive disease (COPD) is a chronic inflammatory lung disease that is highly correlated with cigarette smoking [35]. Exposure to cigarette smoke, air pollutants, or nicotine vapors, sometimes along with genetic predispositions, initiates inflammatory oxidative stress that induces lung parenchyma and alveolar wall damage. The chronic exposition leads to lung emphysema and the development of COPD [36]. Recent studies shed some light on the involvement of EVs in the inflammation process during COPD.

Qiu et al. showed that microparticles (MPs) released by CD4^+^ and CD8^+^ T lymphocytes (TLMPs) were highly elevated in COPD patients. Human bronchial epithelial cells (BECs) treated with TLMP-derived MPs from COPD patients demonstrated increased production of IL-6, monocyte chemoattractant protein 1 (MCP-1), MCP-2, metalloproteinase 9 (MMP-9), and TNF-α, as well as a decreased level of IL-10. These results indicate that TLMP-MPs trigger an inflammatory response in epithelial cells during COPD development [37]. Activated neutrophil-derived exosomes play a role in COPD as well, which was revealed by Ganschmer et al. Exosomes derived from activated neutrophils/polymorphonuclear cells (PMN) had a significantly elevated level of surface neutrophil elastase (NE) and exhibited an ability to directly bind and degrade collagen fibrils. The direct connection was mediated by the macrophage-1 (Mac-1) integrin. Moreover, PMN-derived exosomes caused alveolar enlargement in an in vivo model. The results indicate that neutrophil-derived exosomes contribute to emphysema development during COPD by degrading collagen fibrils in the extracellular matrix (ECM) [38]. The importance of neutrophil-derived MVs was confirmed in a clinical study conducted by Soni et al. The concentration of PMN-derived MVs in BALF was significantly higher in COPD patients. Furthermore, the higher concentrations of PMN-derived MVs correlated with the higher severity of COPD. Interestingly, BALF epithelial cell-derived MVs correlated with the number of exacerbations of COPD [39]. Additionally, alveolar macrophages (AM)-derived MVs may also play a role in COPD progression in smoker patients. The level of BALF-MVs derived from macrophages (CD14^+^) was significantly higher in smokers with COPD than in smokers without COPD and nonsmokers [40].

#### 2.1.3. Cystic Fibrosis

Cystic fibrosis (CF) is an autosomal recessive genetic disease that majorly affects the respiratory system. The mutation in the cystic fibrosis transmembrane conductance regulator (*CFTR*) gene disrupts chloride channels in the apical membrane of epithelial cells which results in the production of dense sputum and cilia dysfunction. A common complication of CF is a recurrent bacterial infection commonly caused by *P. aeruginosa* or *S. aureus.* Moreover, a characteristic feature of CF is neutrophil infiltration in the airways [41]. Remarkably, neutrophils were highly activated after stimulation with EVs that were collected and isolated from a cell line with the *CFTR* mutation. Moreover, these EVs enhanced neutrophil migration and increased their size and granularity. Receptor for advanced glycation end-products (RAGE), phosphor extracellular signal-regulated kinase (ERK), and p38 were elevated in stimulated neutrophils. Additionally, these EVs contained many inflammation-related proteins such as vascular cell adhesion protein (VCAM), epithelial cell adhesion molecule (EPCAM), S100-A12/11, or complement C4. BALF collected from patients with CF revealed higher concentrations of EVs [42]. Rollet-Cohen et al. decided to further investigate the protein cargo of BALF-EVs from CF patients. Proteomic analysis revealed significantly increased levels of neutrophil gelatinase-associated lipocalin (LCN2), superoxide dismutase (SOD2), glutathione peroxidase 3 (GPX3), S100-A12, and synaptosomal-associated protein 23 (SNAP23) [43]. Those proteins are related to mitochondrial metabolism, cancer genesis modulation, and phagocytosis; thus, EVs may play a role in CF progression by modulating various pathophysiological mechanisms [44,45,46].

#### 2.1.4. Asthma

Asthma belongs to the most prevalent pulmonary chronic diseases in developed countries and is a heterogeneous disorder related to chronic airway inflammation, which is extended from upper to peripheral airways and presents different molecular mechanisms [47]. Most of the patients suffer from allergic asthma characterized by a Th2 immune response against harmless environmental stimuli. It involves the production of Th2 cytokines, immunoglobulin E (IgE) production by B cells, and eosinophil recruitment into the airways. There is also a nonallergic Th2 eosinophilic response induced by alarmins and innate lymphoid cells, as well as a non-eosinophilic phenotype driven by Th17/Th1 cells and characterized by neutrophilic inflammation [48]. In any case, the injury of epithelial cells triggered by this inflammation results in characteristic airway remodeling, which includes airway smooth muscle hypertrophy and hyperplasia, mucus hypersecretion, airway narrowing, and hyperresponsiveness to environmental stimuli [49,50].

Basically, EVs from all cell types which are involved in asthma pathology can be found in asthmatic BALF and play a role in the pathological process. It was shown that BALF from asthma patients contains more EVs with higher tetraspanin and HLA-DR expression than in healthy subjects. These EVs contained enzymes for the biosynthesis of leukotrienes, potent proinflammatory mediators involved in asthma. Coculture of bronchial epithelial cells with these EVs increased their leukotriene and IL-8 production, contributing to the inflammation in asthma [51]. Typical asthmatic processes enhance EV secretion by lung epithelial cells, such as IL-13-driven inflammation [52] or bronchoconstriction mechanical stress [53]. It was shown that typical EVs secreted by the dominant cell population in allergic asthma, the eosinophils, promote epithelial cell apoptosis and smooth muscle cell proliferation via TNF or CCL26 [54]. Moreover, they also can act as chemoattractant for eosinophils by increasing nitric oxide (NO) and reactive oxygen species (ROS) levels, and they enhance eosinophil adhesion via intercellular adhesion molecule-1 (ICAM-1) and integrin-α2 upregulation [55]. Similarly, neutrophil-derived EVs enhance neutrophil chemotaxis and adhesion via transcellular metabolite (arachidonic acid) shuttling between neutrophils and platelets. As a result, platelets get activated and induce the endothelial expression of ICAM-1, inducing intravascular crawling and extravasation of neutrophils [56].

#### 2.1.5. Acute Respiratory Distress Syndrome and Acute Lung Injury

Acute lung injury (ALI)/acute respiratory distress syndrome (ARDS) is a devastating respiratory disorder caused by pulmonary edema that is usually developed during excessive inflammation stimulated by infection. The most common clinical conditions associated with ALI/ARDS are bacterial and viral pneumonia, although noninfectious triggers can also lead to acute inflammation. The main pathophysiological mechanisms that contribute to the ALI/ARDS development are a dysregulated inflammation process and increased endothelial and epithelial permeability, resulting in disruption of the microvascular barrier [57]. Various proinflammatory factors take part in the process, including components of EVs.

Recent studies showed a potential role of EVs in ARDS development. The group II secretory phospholipase A2 (sPLA2-IIA) was detected in BALF-EVs obtained from patients with early-stage ARDS, but not in ARDS and non-ARDS patients. Exosomes from early ARDS patients were also enriched with phospholipase A2 group II gene (PLA2G2A) mRNA, whereas exosomes from other patients contained lower concentrations of this mRNA. sPLA2-IIA is a proinflammatory marker, which is able to hydrolyze lung surfactant phospholipids; thus, it can be an important factor that contributes to the initial stage of ARDS progression [58]. Furthermore, BALF-EVs containing miR-466g and miR-466m-5p may play a role in the activation of the nod-like receptor (NLR) family pyrin domain containing 3 (NLRP3) inflammasome and, thus, contribute to the ARDS development. Shikano et al. showed that BALF-EVs enriched with miR-466g and miR-466m-5p activated the NLRP3 inflammasome in bone-marrow-derived macrophages (BMDMs), which resulted in significantly induced pro-IL-1β expression. This may indicate that EVs play a role in the enhancement and dysregulation of the inflammation process during ARDS [59].

ALI may be induced by many infectious triggers including bacterial infection, lipopolysaccharide (LPS), and many more. Zhang et al. aimed to determine whether BALF-MVs collected during lung inflammation caused by LPS and *Klebsiella pneumoniae* contain regulators of the inflammation process. Levels of miR-223 and miR-142 in BALF-MVs and serum-derived MVs were highly elevated compared to the control. MiR-223 and miR-142 showed anti-inflammatory activity by targeting NLRP3 inflammasome components. Furthermore, in vitro studies with BMDMs and THP-1 monocytes after LPS and *K. pneumoniae* stimulation revealed a decrease in intracellular expression of miR-223 and miR-142 while the macrophages were activated. Moreover, MV-miR-223/142 mimics administered in mice pretreated with *K. pneumoniae* revealed decreased macrophage and neutrophil lung infiltration, suppressed inflammasome activity, and lowered expression of IL-1B and IL-18 in lung [60]. Another study conducted by Lee et al. showed that LPS and *P. pneumoniae* induced ALI with elevated levels of BALF-EVs that carried macrophage markers such as CD68. Researchers investigated whether BALF-EVs can promote an inflammatory response in mice lungs. However, BALF-EVs and EV-free BALF collected from an infectious ALI model independently augmented proinflammatory cytokine gene expression, which indicates that not only BALF-EVs but also soluble BALF factors are important for inflammation development. Moreover, BALF-EVs upregulated TLR6 (Toll-like receptor 6), TLR9, CD80, IL-1β, and IL-10 in recipient AMs [61]. Yuan et al. focused on the proinflammatory effect of BALF-EVs obtained from mice with LPS-induced septic lung injury. BALF-EVs from ALI mice had a significantly greater amount of miR-155 and miR-146a. Moreover, researchers confirmed by fluorescence microscopy that BALF-EVs were able to internalize into recipient epithelial cells. Furthermore, lung epithelial cells treated with BALF-EVs obtained from ALI mice overexpressed proinflammatory mediators such as TNF-α and IL-6 and downregulated the tight junction protein (TJP-1) [62]. Additionally, Soni et al. showed that LPS-induced macrophages release MVs that carry a high amount of TNF-αand can activate lung epithelial cells. Moreover, mice treated with LPS-induced macrophage-derived MVs exhibited acute lung injury features such as increased protein and neutrophils levels in BALF, overexpression of ICAM-1 on the surface of both I and II type epithelial cells, and increased concentration of keratinocyte chemoattractant (KC) (mice homolog of IL-8) in BALF [63]. The abovementioned studies indicate that EVs may play various roles during infectious acute lung injury. They may contribute to the reduction of anti-inflammatory factors inside inflammatory cells, resulting in their activation, or transport of proinflammatory factors—mainly from macrophages to the alveolar epithelial cells—which results in overexpression of proinflammatory cytokines and increased alveolar permeability.

Noninfectious triggers such as high tide ventilation, acid exposure, or hyperoxia can also lead to ALI development. Dai et al. investigated the contribution of MVs to ventilation-induced lung injury (VILI). A cargo of MVs obtained from mice with VILI contained significantly more IL-1β, IL-6, and TNF-α. Inhibition of Rho kinase (ROCK), which is a downstream signal of RhoA, yielded attenuation in Limk phosphorylation and MV count reduction. Furthermore, it attenuated the lung inflammation status, suggesting the role of MVs in VILI development [64]. Acid-induced ALI can also be mediated by MVs that are intensively phagocyted by activated residual alveolar macrophages using the tyrosine protein kinase (MerTK) receptor [65]. Moreover, Lee et al. showed that acid-induced lung inflammation upregulated MV release by lung epithelial cells. Epithelial cell-derived MVs and their cargo miR-17 and miR-221 were highly consumed by macrophages. Additionally, MV-containing miR-17 and miR-221 upregulated the expression of integrin B1 and mediated macrophage migration. Furthermore, hyperoxia induced lung epithelial cells to release MVs enriched with miR-320a, miR-221, and miR-342. Those MVs were able to induce macrophage migration by MMP9 upregulation, increased macrophage-derived TNF-α secretion, and activation of nuclear factor kappa-light-chain-enhancer of activated B cells (NF-κB) pathway components. Moreover, the researchers confirmed that those effects were acquired by the MV cargo, miR-221 and miR-320a, which acted synergistically [66]. Another study conducted by Lee et al. showed that noninfectious stimuli such as hyperoxia and acid exposure induced lung inflammation with an increased level of BALF-EVs derived from type I epithelial cells. Hyperoxia-induced epithelial EVs upregulated TLR2, Myd88, TNF-α, and IL-6 expression in AMs but suppressed TLR8 [67]. Taken together, the results indicate that EVs contribute to the noninfectious ALI by transporting proinflammatory cargo mainly from epithelial cells to macrophages, which results in macrophage activation and migration. In summary, EVs can be produced in response to many infectious and noninfectious triggers as shown in Figure 1 and are involved in many processes that induce inflammation during ARDS/ALI development.

#### 2.1.6. Interstitial Lung Disease

Interstitial lung diseases (ILD) is a heterogenous group of respiratory disorders characterized by inflammation and progressive fibrosis of the lung parenchyma [68]. Novelli et al. confirmed that MPs may play a role in ILD development by showing that patients with pulmonary fibrosis exhibited increased levels of MPs. Additionally, MP-associated tissue factor activity was also elevated, which correlated with a reduction in forced vital capacity and diffusion capacity for CO. In vitro experiments showed that oxidative stress increased MP release, and H_2_O_2_ stimulation significantly induced TF-mediated procoagulant activity of MPs [69]. These findings indicate that MPs may contribute to the progression of ILD; however, more studies are needed to confirm this observation and to determine the precise mechanism of MPs action.

#### 2.1.7. Allergic Airway Inflammation

Allergic airway inflammation is a condition characterized by IgE-dependent mast-cell activation, eosinophilic infiltration, and Th2 lymphocyte migration to the airway mucosa [70]. IL-4 and IL-13 are the main cytokines associated with allergic inflammation [71]. Kulshreshtha et al. investigated the effect of epithelial cell-derived exosomes after proallergic Th2 cytokine stimulation. IL-13 was administered to the human bronchial epithelial cell line BEAS-B2 which resulted in increased exosome secretion from epithelial cells. A similar effect was observed after IL-4 and IFN-γ treatment. However, macrophages exhibited a reduction in exosome secretion after treatment with IL-13 and IL-4 but not with IFN-γ. Exosomes released from IL-13-induced epithelial cells increased monocyte proliferation in inflamed lungs of mice. Moreover, the reduced expression of exosomes decreased histopathologic features of allergic inflammation and lowered serum IgE levels and cytokines such as IL-13 [52]. This indicates an important role of exosomes in the development of allergic inflammation. Another study conducted by Shin et al. showed that LPS-induced EVs play a role in the development of an allergen-specific Th1 and Th17 cell response. Their protein cargo increased the effect of sensitization with OVA, which resulted in an escalated accumulation of Th1 and Th17 T cells in the lungs of mice. Moreover, LPS-induced EVs increased the production of IL-6 and TNF-α in lung macrophages. Furthermore, researchers confirmed that the EV action was induced by the protein components. Inhibition of EV release from host cells attenuated the proinflammatory effect [72]. Taken together, EVs may play a role in the induction and progression of allergic airway inflammation. Figure 2 shows the effects of exosomes derived from various type of cells during inflammation processes described in different lung diseases.

#### 2.1.8. Lung Cancer

Lung cancer is a leading cause of death among cancer-related conditions. For clinical use, it is divided into small-cell (SCLC) and non-small-cell lung carcinoma (NSCLC). The latter includes adenocarcinoma, squamous cell, large cell, and bronchoalveolar carcinomas [73]. Lung carcinogenesis is related to various risk factors such as tobacco consumption, genetic susceptibility, diet, air pollution, or chronic infections [74,75]. Many of those factors induce inflammatory processes that are substantial for carcinogenesis, cancer progression, and metastasis. Different leukocyte populations and their ability to release diverse cytokines, chemokines, interleukins, or cytotoxic mediators play a key role in tumor progression [76]. Since EVs are known to transport active molecules such as inflammatory mediators from parental cells into target cells, they may play a role in the cancer progression [77,78].

It was shown that chronic inflammation induced by a lysate of the Gram-negative bacterium nontypeable Haemophilus influenzae (NTHi), often found in the airways of COPD patients, caused a marked increase of exosome levels in BALF of mice. These exosomes demonstrated elevated levels of TNF-α and IL-6 and facilitated the development of lung cancer cells in vitro and in vivo after internalization [79]. Exosomes can also carry a specific cargo from parental cells to reduce its intracellular concentration. Rodriquez et al. showed that tumor cells of NSCLC released exosomes containing miR-126 and miR-144 [15]. MiR-126 is considered to inhibit the proliferation of NSCLC by suppression of the recruitment of mesenchymal cells and inflammatory monocytes, while miR-144 inhibits cancer cell proliferation and progression [15]. A brief overview of the impact of BALF-EVs in inflammation in different lung diseases is given in Table 1.

### 2.2. BALF-EVs Mediate Anti-Inflammatory Effects

Anti-inflammatory effects in the lung have been largely attributed to MSC-derived EVs and have been extensively discussed elsewhere [81,82]. Reported anti-inflammatory effects of EVs from other donor cells in the lung microenvironment are rare. However, emerging evidence demonstrates that EVs play a critical role not only in pathological inflammation but also in normal lung homeostasis, mediating anti-inflammatory crosstalk between the major cellular constituents of the respiratory surface, the respiratory epithelial cells (ECs), and the predominant lung-resident immune cells, the AMs. Here, the AMs are responsible for the tolerogenic phenotype and inflammatory quiescence within the airways and alveolar space [83]. It was shown that these AMs release EVs packed with anti-inflammatory proteins such as suppressors of cytokine signaling SOCS1 and SOCS3, well-known inhibitors of signal transducer and activator of transcription 3 (STAT3) activation, which are then transported to alveolar ECs. They restrain lung inflammation by suppressing the cytokine-induced production of MCP1, a key chemokine responsible for recruitment of monocytes during lung inflammation [84]. BALF collected from patients with lung adenocarcinoma contained lower levels of SOCS3 [80,85]. EVs derived from AMs containing SOCS3 significantly restrained STAT3 activation and reduced proliferation and survival of lung adenocarcinoma cells. Administration of recombinant SOCS3 within synthetic liposomes inhibited proliferation and survival of lung adenocarcinoma cells in vitro, as well as attenuated tumor growth in a lung cancer xenograft model [80].

Researchers from the University of Michigan showed that BALF-EVs may also have anti-inflammatory effects. According to their research, AMs release EVs containing the SOCS3 that attenuate the proinflammatory Janus kinase (JAK)/STAT signaling (activated by cytokines) in BECs. The level of SOCS3-rich EVs was decreased in patients with asthma, as well as in murine asthma models. Results of an in vitro experiment suggested that inflammatory cytokines are responsible for the decrease in SOCS3 secretion and contribute to defects observed in asthma. Interestingly, synthetic liposomes containing SOCS3 had a similar effect to BECs as natural EVs, which opens the possibility of innovative therapy for asthma patients [86]

Another research team proposed a potential method of acute lung injury treatment. As a treatment agent in a murine injury model, exosomes containing a high level of syndecan-1 were used [87]. Syndecan-1, a heparan sulfate proteoglycan, is an important compound of glycocalyx interacting with many agents of disease pathogenesis [88]. Exosomes containing high levels of syndecan-1 were isolated from the medium of lentivirus-transfected mouse pulmonary microvascular endothelial cells (MPMVECs). Syndecan-1 enriched exosomes were proven to decrease the production of pro-inflammatory cytokines (IL-1β, TNF-α, and IL-6), thicken the glycocalyx, and reduce pulmonary edema, which significantly moderated the symptoms of acute lung injury. Furthermore, the anti-inflammatory action of those exosomes was demonstrated in vitro through reducing stress fiber formation and monolayer hyperpermeability primarily stimulated by LPS [87]. Furthermore, it was shown in asthma that airway epithelial cells secrete the lipid phosphatase inositol polyphosphate 4-phosphatase type I A (INPP4A) in the form of extracellular vesicles, which negatively regulate the phosphatidylinositol 3-kinase/serine/threonine protein kinase B (PI3K/Akt) pathway and inhibit airway inflammation and remodeling. Antibody-mediated neutralization of this vesicular INPP4A induced airway hyperresponsiveness, with prominent airway remodeling, subepithelial fibroblast proliferation, and collagen deposition in a murine model of asthma [89]. The anti-inflammatory effects of BALF-EVs are listed in Table 2.

## 3. BALF-EVs as Biomarkers

As it was shown in previous sections, EVs may carry various components (e.g., miRNA, DNA, and proteins) promoting tumor growth or inflammation in many lung diseases. The components, whose concentration/presence is correlated with disease occurrence, may become perfect biomarkers and facilitate early detection of those diseases. BALF seems to be a promising source of EVs as biomarkers of various inflammatory lung diseases and lung cancer, since vesicles are received directly from the disease microenvironment and may appear earlier in BALF than in peripheral circulation. Moreover, the procedure of BALF sampling is not as invasive as tissue obtained during biopsy. In this section, several examples of potential EV-based biomarkers are presented.

### 3.1. Inflammatory Diseases

Recently, scientists from University of Padova analyzed the contribution of microvesicles from BALF to COPD. It turned out that MVs expressing CD14, which plays a key role in the activation of innate immune response cells (e.g., alveolar macrophage) [90], were increased in BALF obtained from smokers with COPD, compared to smokers without COPD and nonsmokers. Moreover, the concentration of CD14^+^ MVs correlated positively with pack-years of smoking and negatively with the lung function expressed as lung function parameter (FEV); 1% predicted [40].

Lee et al. investigated subpopulations of miRNA-rich EVs isolated from mice BALF. Research carried out on mice showed that miRNA is delivered by EVs to alveolar macrophages and may activate the proinflammatory genes *IL-1β* and macrophage inflammatory protein 2 (*MIP2*) during *Pseudomonas aeruginosa* pneumonia. Moreover, miRNA participated in inducing innate immune response by promotion of inflammasome activation, neutrophil recruitment, and M1-macrophage polarization [91]. Although these miRNA-rich EVs play such a crucial role in mediating inflammation during the studied bacterial lung infection, it turned out that they constitute only about 9% of all BALF-EVs and are mostly derived from alveolar epithelial type-I cells (ATIs). Because caveolin-1, a lipid raft protein, is a well-known ATI cell marker, miRNA-rich EVs derived by ATIs are also enriched in this protein. As a result, caveolin-1 may become a biomarker of inflammation in bacterial lung infection [92].

Lately, nucleic acids, for example, miRNA, have been increasingly discovered as potential biomarkers in various diseases. This situation may happen due to the more and more available high-throughput techniques, such as next-generation sequencing (NGS) or microarrays. Using those high-throughput methods, Liu et al. generated a list of miRNAs from BALF as potential biomarkers of idiopathic pulmonary fibrosis (IPF). A quantitative assay using microarray showed that miR-125b, miR-128, miR-21, miR-100, miR-140-3p, and miR-374b were upregulated, while let-7d, miR-103, miR-26, and miR-30a-5p were downregulated in patients with IPF. miR-30a-5p was chosen for further analysis using RT-qPCR, whereby a significant difference in expression level between IPF patients and healthy controls was confirmed. Further investigations showed that miR-30a-5p decreased the expression of transforming growth factor β (TGF-β)-activated kinase 1/ mitogen-activated protein kinase kinase kinase 7 (MAP3K7) binding protein 3 (TAB3). As a result, the decreased level of miR-30a-5p caused increased TAB3 expression, which can be crucial for IPF progression [93].

A similar study was executed for asthma patients, where expression of miRNA from BALF-EVs was analyzed using microarrays and nanoscale qPCR. Due to high-throughput quantitative analysis, the expression level of 24 miRNAs was identified as being significantly different in asthma patients compared to healthy controls. Although the asthma patients in this study had a mild intermittent stable disease, expression of these miRNAs was highly correlated with the FEV1 parameter. Pathway analysis gave information about their influence on cytokines of known importance in asthma, including IL-13, IL-10, IL-6, and IL-8, as well as on the MAPK and JAK/STAT signaling pathways. Lastly, application of multivariate models enabled separation of asthma patients and healthy controls (by choosing a subset of 16 miRNAs) with a predictive power of 73% [94].

Expression analysis of miRNAs from BALF-EVs was also carried out for patients with pulmonary sarcoidosis at different stages of illness. Analysis using qPCR showed that miR-146a and miR-150 from EVs were upregulated in chest X-ray stage II (CRX-II) compared to CRX-I. Considering that there are four stages of sarcoidosis severity (CRX-I–IV), chosen miRNAs may be promising biomarkers in the early stages of the illness. Moreover, miR-146a and miR-150 expression was correlated with pulmonary function, i.e., vital capacity (VC) and FEV1/forced vital capacity (FVC) ratio [95].

As biomarkers in lung diseases, EVs derived from specific cell types can also be used. Researchers from Marseille observed that patients with ARDS had a significantly higher concentration of BALF microparticles derived from leucocytes (LeuMP) than spontaneous breathing (SB) and ventilated control (VC) groups. Interestingly, the concentration of LeuMP in BAL, measured on the third day of ARDS, was significantly higher for survivors. Lower levels of LeuMP for nonsurvivors resulted in a decreasing inflammatory response at early stages of the illness, which indicates a protective role of these EVs in ARDS [96].

### 3.2. Lung Cancer

Similarly, to the presented cases of inflammatory diseases, EV-derived miRNAs also have huge potential as biomarkers for lung cancer. Researchers from Pusan National University found two miRNAs, miR-126 and Let-7a, whose expression levels in BALF-EVs of adenocarcinoma patients were significantly increased compared to healthy controls. Moreover, these miRNAs were detected in patients at an early stage, which is crucial for the curability of lung cancer. A higher expression of miR-126 was found also in cancer tissue compared to normal lung tissue, which confirms that the EV miRNA signature may be a representative sample of the tumor microenvironment [97].

Worth mentioning is one more high-throughput technique, mass spectroscopy (MS), which has been widely applied in biological research in recent times. Proteomic analysis of BALF from lung cancer and noncancer patients using liquid chromatography MS (LC–MS) showed EVs as the most relatively protein-enriched cellular component compared with other compartments such as the cytosol, nucleus, extracellular space, and plasma membrane. Moreover, 35% of proteins with significantly different levels for lung cancer and control patients were classified as EV proteins [98], which proclaims them as a promising source of new biomarkers. Another study using LC–MS displayed significant upregulation of proteins involved in the ERK and nicotinamide adenine dinucleotide phosphate (NADP) binding pathways in BALF-EVs (similarly as in tumor tissue) from lung cancer patients. Both pathways were described as involved in cancer promotion [99,100]. Researchers selected also the protein DNA (cytosine-5)-methyltransferase 3β (DNMT3B) complex as significantly upregulated in BALF-EVs from lung cancer patients [101]. The DNMT3B complex is responsible for de novo epigenetic modifications and considered as a prospective therapeutic target [102].

Lastly, the use of target-specific drugs, e.g., epidermal growth factor receptor tyrosine kinase inhibitors (EGFR-TKIs), decreased the mortality of lung cancer patients. Nevertheless, fast and precise identification of patients which may benefit from these kinds of targeted treatment is very important. Currently, EGFR genotyping is handled by tissue biopsy or alternatively by liquid biopsy using cell-free DNA (cfDNA) [103]. According to the study performed by Hur et al., EGFR genotyping using DNA from BALF-EVs had a 100% correspondence to tissue typing, while the procedure using cfDNA had in comparison a sensitivity of only 71.6% [16]. These results form the basis for replacement of tissue biopsy with BALF-EVs as a source of DNA for genotyping, which makes the procedure less invasive.

In the case of cancers, EVs can be also applied as biomarkers of drug resistance. Icotinib, an EGFR-TKI, is commonly used for the treatment of NSCLC patients. However, acquired drug resistance drastically limits its clinical application. Yu et al. analyzed 10 specific mRNAs from BALF-EVs in NSCLC patients after treatment with icotinib. One mRNA of those 10, which is a product of the proto-oncogene *MET*, reported to be involved in EGFR-TKIs resistance [104], was only expressed in some patients. It turned out that *MET* mRNA was present only in EVs from patients whose treatment was ineffective, and where metastasis was observed. Moreover, in vitro experiments suggested that EVs containing *MET* mRNA trigger activation of migration and invasion ability in icotinib-nonresistant cells [105]. The discussed usage of BALF-EVs as biomarkers of diverse lung diseases is summarized in Table 3.

Taking everything into account, BALF-EVs carry many various molecules including proteins, DNA, or RNA, which can be applied as biomarkers. As a result, BALF-EVs have huge potential for diagnostic purposes and may contribute to the recognition of lung cancer at an early stage of illness. The great potential of EVs in diagnostics is supported by the fact that EVs isolated from blood are acknowledged as an important complement of the classical types of liquid biopsy such as cell-free DNA (cfDNA) and circulating tumor cells (CTC), which are currently being used in diagnostics commercially. Yu et al. suggested that a combination of liquid biopsy modalities (cfDNA, EVs) may improve the sensitivity in comparison to single markers, and the same may become true for BALF EVs [106]. Recently, it was demonstrated that BALF-EV-DNA as a liquid biopsy for *EGFR* genotyping is especially sensitive compared to plasma cfDNAs, especially in detecting pT790M mutations in acquired resistance patients [16]. In a subsequent study, Hur et al. showed that BALF EV-based *EGFR* genotyping has an even better mutation detection rate than tissue/cytology-based typing and that its sensitivity increases with tumor stage, enabling researchers to follow the disease progression and potentially the response to treatment without burdensome repeat biopsies [107]. Lastly, an opportunity for the identification of lung cancer by BALF-derived liquid biopsy of cfDNA and tumor cells emerged [108,109]. Although research on the clinical usefulness of different components of a liquid biopsy from BALF is only just developing and is nowhere as advanced as similar research concerning blood, it is for sure reasonable and promising, since BALF enables the gathering of both cellular (CTCs) and noncellular components (cfDNA, EVs) directly from the tumor microenvironment, and it provides increased diagnostic sensitivity when compared to blood.

## 4. Conclusions

As evident from the many studies described above, BALF-EVs are important vehicles of cellular communication and modulators of pathological conditions, and they have begun to enter research as potential diagnostic and/or prognostic biomarkers for pulmonary disorders. Although plasma and serum are the most popular sources of EVs as they are the most easily accessible, in the case of lung diseases, BALF seems to be a better source, mainly because of its direct proximity to the lung microenvironment. Cytological examinations of BALF have already been performed for many years during the diagnosis of diffuse parenchymal lung disorders, interstitial lung diseases, and infections, and they are now being correspondingly applied for EV research. Another advantage of BALF as an EV source in comparison to blood is the lower complexity of the biological fluid, which significantly expedites EV isolation. Due to the high content of proteins and lipoproteins in plasma or serum, the separation of EVs from these sources poses a real challenge, and the currently available isolation methods offer EV samples of usually unsatisfactory purity and yield.

The translation of BALF-EV research to the clinic progresses very slowly. To date, most EV-related clinical trials were designed for lung cancer to detect genetic mutations and programmed cell death ligand 1 (PD-L1) expression in combination with targeted therapy or immunotherapy, and they were primarily based on blood-derived EVs [110]. However, encouraged by the promising results of BALF-EV genotyping studies described above, the first clinical attempts to replace blood EVs with BALF-EVs as a source of DNA are being undertaken, e.g., in the olmutinib phase 2 clinical trial where T790M mutations in NSCLC patients are being detected on the basis of BALF-EV DNA [111], and more similar trials are planned. Among the potential EV-based biomarkers, EV miRNAs including BALF are the most studied and are considered to be promising biomarkers, especially for lung cancer. Although preclinical research accumulated convincing evidence about the diagnostic and prognostic potential of BALF-EV miRNAs, corresponding clinical trials are still in a very early stage of development. The reason is that BALF-EV research and its translation to the clinic are still facing technical issues and unresolved scientific problems. The relatively low EV concentration in BALF requires the development of an effective high-yield isolation method, and uniform sample collection procedures should be established to exclude experimental bias and ensure inter-laboratory reproducibility. It has to be clarified if the alterations in EV content and functions identified in pulmonary disorders are a cause or a consequence of the disease. Furthermore, since BALF offers a mixture of EVs from different donor cells, suitable techniques to differentiate the EV subpopulations have to be developed and implemented, e.g., the immune capture of specific EVs or direct EV phenotyping by nanoflow cytometry or nanoparticle tracking analysis. In vitro-based studies on cell cultures, which oversimplify the complex EV interactions in the real organism, are needed to be complemented by in vivo models and large-scale studies of patient-derived samples, which would more closely mirror the complicated reciprocal signaling network between EVs and cells in the lung microenvironment. They may also help to answer whether the EV-mediated signaling within the lung microenvironment is targeted, which molecules are responsible for the transfer to the donor cells, what are the mechanistic steps involved in the transmission of vesicular information to donor cells, and how they may be modulated by factors in the lung microenvironment. In addition to these basic science obstacles and questions that have to be overcome and answered, more large-scale prospective multi-institutional studies are needed to decipher the clinical significance of BALF-EV cargo. Learning from the experience of blood EV studies, whole panels of miRNAs or proteins should be evaluated instead of single markers to ensure high specificity and sensitivity. Another important issue to consider in future clinical implementations is the fact that pulmonary diseases are physiologically complex and occur at many stages of severity, and future translational research on BALF-EVs as biomarkers should take into consideration these differences. A better understanding of the interactions of BALF-EVS with cells within the inflammatory lung microenvironment is also necessary for a future therapeutical application of EVs, e.g., MSC-derived EVs, for the treatment of lung diseases. Lastly, in-depth profiling of distinct EV populations in BALF including proteomics, RNA-seq, lipidomics, and metabolomics will elucidate their role in the pathogenesis of inflammatory lung diseases and lung cancer, as well as provide a novel source of specific and minimally invasive biomarkers and treatment strategies.

## Figures and Tables

**Figure 1 ijms-22-03651-f001:**
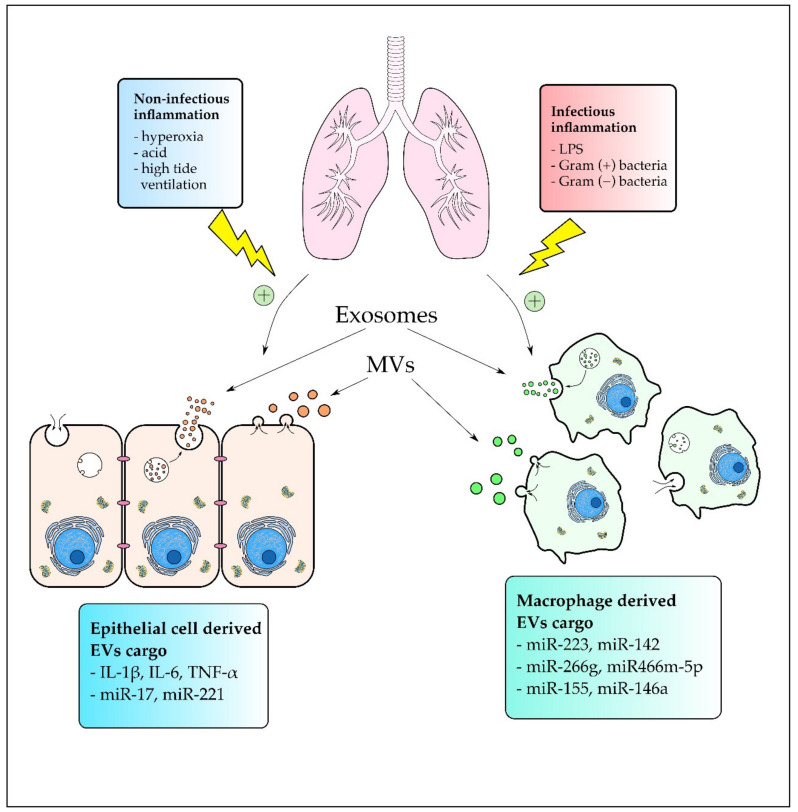
The effect of infectious and noninfectious stimuli on extracellular vesicle (EV) release during acute lung injury (ALI)/acute respiratory distress syndrome (ARDS) development. Noninfectious stimuli mainly induce EV production and release by alveolar epithelial cells, whereas infectious stimuli were correlated mainly with alveolar macrophage (AM) activation and AM-EV excretion.

**Figure 2 ijms-22-03651-f002:**
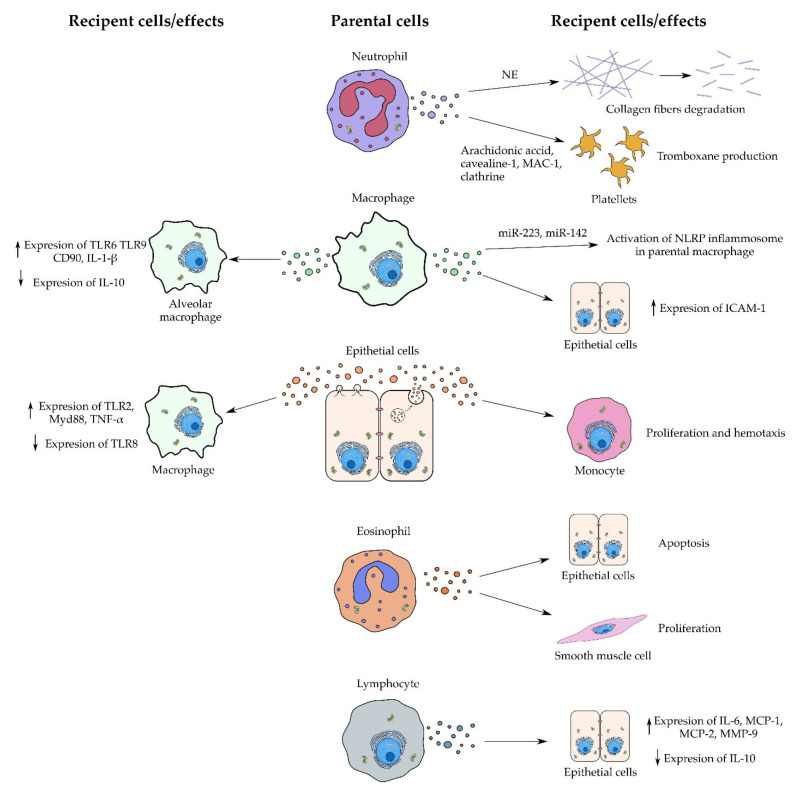
Action of EVs derived from various types of parental cell during the inflammation process. EVs can act indirectly by inducing parental cells and directly as shown in the case of collagen fiber degradation.

**Table 1 ijms-22-03651-t001:** Impact of EVs found in bronchoalveolar lavage fluid (BALF) in different lung diseases on inflammation.

Disease	EV Source	Cargo	Effects	Recipient Cells	Infl.	Ref.
**Sarcoidosis**	BALF	NRG1	↑ IFN-γ, IL-13 in autologous PBMCs and IL-8 in ECs	PBMCs and epithelial cells	↑	[30]
	BALF	-	↑ IL-1β, IL-6, TNF, and CCL2	Monocytes, PBMCs	↑	[33]
**Chronic obstructive pulmonary disease (COPD)**	T-lymph CD4^+^/CD8^+^	-	↑ IL-6, MCP-1, MCP-2, MMP-9, and ↓ IL-10 in BECs	BECs	↑	[37]
	Neutrophils	Neutrophil elastase	Degradation of collagen fibers	-(DIRECT activity)	↑	[38]
	Neutrophils	-	Correlation with COPD severity	-	↑	[39]
	AMs	-	Correlation with COPD severity in smokers	-	↑	[40]
**Asthma**	BECs	-	Proliferation and chemotaxis of monocytes	Monocytes, macrophages	↑	[52]
	Eosinophils	-	Apoptosis of BECs, ↑ of CCL26, TNF, POSTN; proliferation of smooth muscle cells, ↑ CCR3, VEGFA	BECs, smooth muscle cells	↑	[54,55]
	Neutrophils	Arachidonic acid, caveolin-1, Mac-1, clathrin	Thromboxane production in platelets, ICAM-1 expression in endothelial cells, neutrophil recruitment	Platelets	↑	[56]
	BALF	Enzymes for leukotriene production	Leukotriene and IL-8 production in BEC	BECs	↑	[51]
**Acute** **respiratory distress syndrom (ARDS)/** **Acute lung injury (ALI)**	BALF	sPLA2-IIA	Hydrolysis of lung surfactant phospholipids and ↑ inflammation in early stage of ARDS development	-	↑	[58]
	BALF	miR-466g, miR-466m 5p	activation of NLRP3 inflammasome, ↑ expression of pro-IL-1β in BMDMs	Macrophages	↑	[59]
	Macrophages	miR-223, miR-142	attenuation of inhibitory effect on NLRP3 inflammasome by ↓ miR-223 and miR-142 intracellular concentrations	-	↑	[60]
	BALF	IL-1β, IL-6, TNF-α	↑ Cell counts, protein concentration and IL-1β in BALF	-	↑	[64]
	AECs	-	↑ TLR2, Myd88, TNF-αand ↓ TLR8 expression in AMs	AMs	↑	[61]
	Macrophages	-	↑ TLR6, TLR9, CD80, IL-1β and IL-10 expression in alveolar macrophages	AMs	↑	
	BALF	miR-155, miR-146a	↑ TNF-αand IL-6 and attenuation of ZO-1 protein expression	ECs	↑	[62]
	BALF	-	↑ IL-6 and KC (the murine IL-8 homolog)	ECs	↑	[65]
	Macrophages	TNF-α	Activation of LECs and ↑ ICAM-1 on their surface, ↑ protein concentration, neutrophil count and KC in BALF	ECs	↑	[63]
	ECs	miR-17, miR-221	↑ integrin B1 expression, ↑ macrophage migration	Macrophages	↑	[66]
**Interstitial lung** **disease**	-	-	Procoagulant tissue factor activity	-	-	[69]
	ECs	-	↑ Monocyte proliferation	Monocytes	↑	[52]
	-	-	Th1 and Th17 ↑ lung accumulation, ↑ macrophage IL-6 and TNF-alfa production	Th1 and Th17 lymphocytes, macrophages	↑	[72]
**Cystic** **fibrosis**	CF BECs (CFBE41o)	VCAM, EPCAM, S100-A12/11, complement C4	↑ Neutrophil activation and migration	Neutrophils	↑	[42]
	BALF	LCN2, SOD2, GPX3, S100-A12, SNAP23	-	-	-	[43]
**Lung** **cancer**	AMs	SOCS3	STAT3 inhibition in IL-6-induced adenocarcinoma cells	Lung adenocarcinoma	↓	[80]
	NSCLC	miR-126, miR-144	↓ miR-126 and miR-144 intracellular concentrations in NSCLC led to ↑ cancer proliferation, progression, and metastasis	-	↑	[15]

↑ increase, ↓ decrease, - not specified, Infl.: effect on inflammation, Ref.: reference

**Table 2 ijms-22-03651-t002:** Anti-inflammatory effects of EVs in different lung diseases.

EV Source	Recipient Cells	Anti-Inflammation Activity	Ref.
**AMs**	Alveolar and airway ECs	AM-derived vesicular SOCS3 inhibits STAT signaling in BEC allergic inflammation. Homeostatic secretion of SOCS3 by AMs is impaired in animal models of allergic asthma. Intrapulmonary treatment with SOCS3 liposomes abrogates both the cellular and the molecular components of allergic inflammation.	[86]
**Mouse pulmonary microvascular endothelial cells (MPMVECs)**		Syndecan-1-positive exosomes reduce expression of proinflammatory cytokines, such as IL-1β, TNF-α, and IL-6 following LPS challenge, and reduce stress fiber formation and monolayer hyperpermeability in vitro, as well as pulmonary edema in acute lung injury in vivo	[87]
**AMs**	Alveolar and airway ECs	Exosomal SOCS1 and SOCS3 within MVs are internalized by alveolar ECs and suppress cytokine-induced JAK/STAT signaling downregulating the production of the monocyte chemoattractant protein 1 (MCP-1), one of the key chemokines responsible for recruitment of monocytes to the lung during inflammation	[84]
**Airway ECs**		INPDRa^+^ EVs inhibit airway inflammation and remodeling in a murine model of asthma	[89]
**AMs**	ECs in lung cancer	EVs containing SOCS3 inhibit STAT3 activation, proliferation, and survival of lung adenocarcinoma cells. SOCS3-EV levels are low in NSCLC patients and in a murine lung cancer model. Intratumoral injection of SOCS3 liposomes attenuated tumor growth in a murine xenograft model.	[80]

**Table 3 ijms-22-03651-t003:** BALF-EVs as biomarkers of different lung diseases.

Marker Type	Disease	Marker	Significant Difference/Feature	Ref.
**Protein**	COPD	CD14	MVs CD14^+^ ↑ in BALF, significant correlation with packyears and FEV1%	[40]
	Bacterial lung infection	Caveolin-1	Caveolin-1, component of miRNA-rich EVs from lung epithelial type-I cells, dramatically ↑ during development of acute lung injury	[92]
	ARDS	CD45	LeuMP (CD45^+^) ↑ in survivors on third day of illness	[96]
	Lung cancer	DNMT3B complex	DNMT2B complex ↑ in lung cancer patients	[101]
**miRNA**	Idiopathic pulmonary fibrosis	miRNA (miR-125b, miR-128, miR-21, miR-100, miR-140-3p, miR-374b, let-7d, miR-103, miR-26 and miR-30a-5p)	miR-125b, miR-128, miR-21, miR-100, miR-140-3p, miR-374b ↑, let-7d, miR-103, miR-26 and miR-30a-5p ↓	[93]
	Sarcoidosis	miRNA (miR-146a and miR-150)	miR-146a and miR-150 ↑ in CRX-II compared with CRX-I	[95]
	Asthma	miRNA	Set of 16 miRNAs significantly altered in asthmatic patients compared with healthy subjects	[94]
	Lung adenocarcinoma	miR-126 and Let-7a	miR-126 and let-7a levels are significantly ↑ in BALF from early-stage lung adenocarcinoma patients than control	[97]
**DNA**	NSCLC	EV-derived DNA	EGFR genotyping of BALF-EVs enables detection of mutation with 100% correspondence to tissue typing	[16]
**mRNA**		*MET* mRNA	*MET* mRNA present in exosomes isolated from BALF of icotinib-resistant patients	[105]

## Data Availability

Not applicable.

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
