# Peer review of "EVs from BALF—Mediators of Inflammation and Potential Biomarkers in Lung Diseases"

_ijms, 2021, doi:10.3390/ijms22073651_

Round 1

Reviewer 1 Report

  1. Based on my understanding for the review content, I would recommend that the authors consider to change the title of this manuscript.
  2. However obvious you think it is, please define every abbreviation on its first mention in every section in the text.
  3. I don't get the aim of reviewing unpublished results in section 1.2, lines 65-70, except if you as an author plan to ruin the novelty of your future research manuscript. Do consider publish these results before this review if you insist (owing to their important discussions) to mention them in this manuscript.
  4. I suggest that the authors introduce a schematic paragraph after the introduction to illustrate the different scientific outcomes of every following section
  5. Section 2.1.2 is very long and could have been reduced its size without loosing its content. Authors should reconsider greatly the length of this part in relation to the more relevant part section 3, which did not receive the same depth, although it is more interesting regarding the applications of BALF-EVs.
  6. The conclusion does not really cover all the aspects discussed in this review manuscript and more relevant discussion should be included with a separate paragraph for the potentials and possible future directions regarding research on BALF-EVs.

Reviewer 2 Report

Lukasz Zareba and the co-authors have shown a very interesting study of “EVs from BALF – mediators of inflammation in lung diseases and lung cancer”. Different from the conventional liquid biopsy research, they mainly focus on the studies about EVs from bronchoalveolar lavage fluid rather than blood samples. Circulating extracellular vesicles is an important focus from liquid biopsy application and has a huge potential to be a powerful tool for cancer diagnosis. This review summarizes the role of EVs in various lung diseases, providing a novel perspective for further exploration of the clinical relevance and mechanism of EVs.

However, there are still some major points that need to be further improved in the manuscript.

  1. The methods of EVs isolation from BALF, line 88-92.

The number of researches on BALF-EVs is relatively small. Thus, it might be necessary to make a more specific introduction to the EV- isolation methods. Comparing the advantage and feasibility of several ways is important for the study.

  1. The comparison of EVs quality between two different biological liquid origin, Line 53-72.

The author elaborated on the data that there are more EVs isolated from plasma than BALF, which is consistent with the results of other studies. However, the explanation of BALF-EVs shows a higher quality than plasma-EVs is lack persuasiveness to some extent, and it is recommended to add stronger evidence of the advantage of EVs from the BALF.

  1. The format of all the tables in the manuscript is not common, and it is suggested to change to a three-line chart.
  2. The clinical application prospects of BALF-derived EVs should be discussed in depth in the manuscript, for instance, what are the key prospects (inflammatory diseases or cancer) and solution of challenges for the future clinical application of EVs-derived markers for lung disease patients?
  3. Cell-free DNA (cfDNA) and circulating tumor cell (CTC) also important roles in liquid biopsy, some studies have demonstrated the benefit to improve the diagnosis with the combination of different components (EVs, cfDNA and CTC) in blood samples. Is it feasible for performing similar research in BALF?
  4. The contribution of each author in the manuscript is missing, better to update with a clear explanation.

Round 2

Reviewer 1 Report

The manuscript has been improved a lot.

However, in line 136 to 141,where authors descrthe content of section 2 and 3,they mistakely described section 2 as second paragraph and section 3 as third paragraph. So please replace the word (paragraph) by the word (SECTION).

The tile is much better suiting the manuscript. However isn't lung cancer is a lung disease? You may omit the phrase (Lung cancer) from title and it is anyways mentioned in the keywords. 

Author Response

We replaced the word paragraph by the word section.

Following the reviewer's suggestion we omitted the phrase lung cancer in the title.

Thank you very much for your valuable suggestions that helped us to improve the manuscript!

Reviewer 2 Report

The quality and the context have been improved. The resolution of the figures could be optimised as well. In the end, it is an interesting work sharing with scientific community

Author Response

We increased the resolution of the figures according to the suggestion.

Thank you very much for your valuable suggestions that helped us to improve the manusript!